# Development and Validation of a Model to Predict Severe Hospital-Acquired Acute Kidney Injury in Non-Critically Ill Patients

**DOI:** 10.3390/jcm10173959

**Published:** 2021-08-31

**Authors:** Jacqueline Del Carpio, Maria Paz Marco, Maria Luisa Martin, Natalia Ramos, Judith de la Torre, Joana Prat, Maria J. Torres, Bruno Montoro, Mercedes Ibarz, Silvia Pico, Gloria Falcon, Marina Canales, Elisard Huertas, Iñaki Romero, Nacho Nieto, Ricard Gavaldà, Alfons Segarra

**Affiliations:** 1Department of Nephrology, Arnau de Vilanova University Hospital, 25198 Lleida, Spain; mpmarcom.lleida.ics@gencat.cat (M.P.M.); mlmartin.lleida.ics@gencat.cat (M.L.M.); asegarra.lleida.ics@gencat.cat (A.S.); 2Department of Medicine, Autonomous University of Barcelona, 08193 Barcelona, Spain; 3Institute of Biomedical Research (IRBLleida), 25198 Lleida, Spain; mibarz.lleida.ics@gencat.cat (M.I.); spico.lleida.ics@gencat.cat (S.P.); 4Department of Nephrology, Vall d’Hebron University Hospital, 08035 Barcelona, Spain; nramos@vhebron.net (N.R.); jdelatorre@althaia.cat (J.d.l.T.); 5Department of Nephrology, Althaia Foundation, 08243 Manresa, Spain; 6Department of Informatics, Vall d’Hebron University Hospital, 08035 Barcelona, Spain; jprat@parcdesalutmar.cat (J.P.); mtorresfr@bellvitgehospital.cat (M.J.T.); nnieto@gencat.cat (N.N.); 7Department of Development, Parc Salut Hospital, 08019 Barcelona, Spain; 8Department of Information, Southern Metropolitan Territorial Management, 08028 Barcelona, Spain; 9Department of Hospital Pharmacy, Vall d’Hebron University Hospital, 08035 Barcelona, Spain; bmontoro@vhebron.net; 10Laboratory Department, Arnau de Vilanova University Hospital, 25198 Lleida, Spain; 11Technical Secretary and Territorial Management of Lleida-Pirineus, 25198 Lleida, Spain; gfalcon.lleida.ics@gencat.cat (G.F.); mcanales.lleida.ics@gencat.cat (M.C.); 12Informatic Unit of the Catalonian Institute of Health—Territorial Management, 25198 Lleida, Spain; ehuertas.lleida.ics@gencat.cat; 13Territorial Management Information Systems, Catalonian Institute of Health, 25198 Lleida, Spain; iromero.lleida.ics@gencat.cat; 14Amalfi Analytics S.A, 08018 Barcelona, Spain; ricard@amalfianalytics.com

**Keywords:** acute kidney injury, hospital-acquired, electronic health data records, risk score

## Abstract

Background. The current models developed to predict hospital-acquired AKI (HA-AKI) in non-critically ill fail to identify the patients at risk of severe HA-AKI stage 3. Objective. To develop and externally validate a model to predict the individual probability of developing HA-AKI stage 3 through the integration of electronic health databases. Methods. Study set: 165,893 non-critically ill hospitalized patients. Using stepwise logistic regression analyses, including demography, chronic comorbidities, and exposure to risk factors prior to AKI detection, we developed a multivariate model to predict HA-AKI stage 3. This model was then externally validated in 43,569 non-critical patients admitted to the validation center. Results. The incidence of HA-AKI stage 3 in the study set was 0.6%. Among chronic comorbidities, the highest odds ratios were conferred by ischemic heart disease, ischemic cerebrovascular disease, chronic congestive heart failure, chronic obstructive pulmonary disease, chronic kidney disease and liver disease. Among acute complications, the highest odd ratios were associated with acute respiratory failure, major surgery and exposure to nephrotoxic drugs. The model showed an AUC of 0.906 (95% CI 0.904 to 0.908), a sensitivity of 89.1 (95% CI 87.0–91.0) and a specificity of 80.5 (95% CI 80.2–80.7) to predict HA-AKI stage 3, but tended to overestimate the risk at low-risk categories with an adequate goodness-of-fit for all risk categories (Chi^2^: 16.4, *p*: 0.034). In the validation set, incidence of HA-AKI stage 3 was 0.62%. The model showed an AUC of 0.861 (95% CI 0.859–0.863), a sensitivity of 83.0 (95% CI 80.5–85.3) and a specificity of 76.5 (95% CI 76.2–76.8) to predict HA-AKI stage 3 with an adequate goodness of fit for all risk categories (Chi^2^: 15.42, *p*: 0.052). Conclusions. Our study provides a model that can be used in clinical practice to obtain an accurate dynamic assessment of the individual risk of HA-AKI stage 3 along the hospital stay period in non-critically ill patients.

## 1. Introduction

Acute kidney injury (AKI) is a global concern with a high incidence among hospitalized patients [1,2]. The incidence of hospital-acquired AKI (HA-AKI) ranges between 5 and 15% or 30–45 cases/1000 hospital admissions/year but shows an increasing trend as hospitalized patients are older and subjected to more interventional diagnostic and treatment techniques, and exposed to the effects of nephrotoxic drugs [3,4,5]. In addition, AKI has been associated with significant increases in health care resource utilization and costs in patients who are hospitalized, and with long-term morbidity and mortality after hospital discharge [6,7,8,9,10,11]. Numerous studies on AKI have been published in patients admitted to intensive care units, in which the causes, risk factors, mortality, and the influence of different treatment strategies have been identified [12,13,14,15,16,17]. The epidemiology of acute renal failure in patients admitted to conventional hospitalization wards is much less known [18]. Since a large part of the AKI episodes are due to potentially avoidable causes, knowing as accurately as possible the individual risk at any time of hospital stay could help decision making and implementation of preventive measures to reduce the incidence of hospital AKI [19,20]. The diagnostic approach to in-hospital AKI has undergone a significant change over time. The old detection models were based on the communication of the cases at the time of the diagnosis, by conventional analytical controls, and were subject to the influence of multiple sources of error that motivated avoidable delays in the identification of cases and in the adoption of treatment measures [21]. With the appearance of electronic laboratory data records, electronic alert systems were designed. These systems allow the detection of all cases at an early stage, but they do not allow to adopt preventive measures since they detect the problem once it has occurred [22]. The evolution of the management systems of the in-hospital AKI has gone in the direction of the development of predictive models of individual risk, whose purpose is to be able to anticipate the episode of AKI and to carry out prevention measures appropriate to the particular situation of each patient [23]. In recent years, several models have been developed and validated to allow the estimation of the risk of suffering AKI during hospitalization, but the results of early diagnosis and intensive interventions in terms of reduction of morbidity and mortality have been discordant and inconclusive [24]. The studies analyzing the epidemiology and risk factors associated with AKI in non-critically ill patients have two main limitations to identify accurately the risk factors associated with HA-AKI. First, most of them are based on demographic characteristics and comorbidities that have been registered retrospectively, from the discharge administrative codes, and therefore, are subject to a potential bias in the collection of coded information [25]. Secondly, they do not allow to know whether the exposure to risk factors preceded or not the detection of the AKI episode [26]. Thirdly, they do not allow to identify the categories of severe AKI. Wu L. et al. recently published an article where the risk factors that predict the presentation of severe AKI were defined, but they included both ICU and non-ICU patients and no external validation was performed [27]. Our group recently developed a model that overcame some of those limitations and provides an accurate dynamic assessment of the individual risk of suffering AKI along the whole hospital stay period in patients admitted into non-critical hospitalization wards [28]. However, although this model allows AKI to be accurately predicted, because of a lack of statistical power, it does not allow to detect the risk of developing AKI-3 severity stage, which is the one associated with greater morbidity, related to the severity of complications and, in many cases, to the need for replacement of kidney function. The aim of our study was to develop and externally validate a model to predict the risk of HA-AKI stage 3 in hospital-acquired AKI in non-critically ill patients.

## 2. Methods

This study was performed at two different hospital centers. The first center developed the predictive model (study set) and the second center performed the external validation of the predictive model (validation set).

### 2.1. Study Set

The study set included patients admitted to the Vall d’Hebron hospital from January 2011 to December 2017. Vall d’Hebron is a tertiary hospital that provides assistance to a population of 500,000 habitants in Barcelona, Spain, and develops all kinds of medical and surgical procedures, including neurosurgery, cardiac surgery, endovascular catheter-guided procedures as well as lung, liver, kidney and bone marrow transplantation programs. We included all patients >18 years of age who were admitted to hospital along this period and did not meet any of the following exclusion criteria: 1.- admission for community-acquired AKI, 2.- hospital stay < 24 h, 3.- admission for elective heart surgery, 4.- direct admission from the emergency room to the intensive care units (ICUs), 5.- admission as a recipient of renal, lung, liver or bone marrow transplant, 6.- absence of serum creatinine measurements done at least 12 months after hospital admission, 7.- chronic hemodialysis treatment and 8.- denial to give a written consent to participate in the study. Community-acquired AKI was diagnosed whenever patients met the AKI criteria within the first 24 h of hospital admission. Patients initially admitted to conventional hospitalization wards who afterwards required admission into ICUs were only included if the AKI episode was detected while they were admitted in non-critically ill wards, prior to their admission into the ICUs. 

### 2.2. Baseline Kidney Function

Our patient care system integrates the laboratory databases of the hospital and primary care registers, thus allowing historical data to be obtained for all patients who are hospitalized, provided that these data appear in those registers. Baseline kidney function was obtained from the electronic laboratory data records of primary health care and defined as the most recent glomerular filtration rate, estimated by the CKD-EPI equation, within the 12 months prior to hospital admission. 

### 2.3. Definition of AKI Severe

AKI was defined and classified in severity stages according to the KDIGO (Kidney Disease Improving Global Outcomes) clinical practice guidelines [29]. Severe AKI HA-AKI was defined as an increase serum creatinine of at least ×3 over the baseline or ≥4 mg/dL, occurring from the first 24 h to any time within hospital admission.

### 2.4. AKI Detection

A software integrated into the electronic laboratory database was used to perform repeated comparisons among all serum creatinine levels available for each patient during hospital stay and generated an identification code, assigning a 1 when the HA-AKI stage 3 criteria were met and a 0 when not. The date of HA-AKI stage 3 detection was also recorded. The number of the admission episode was used as a filter so that patients with more than one HA-AKI stage 3 episode during hospital stay were entered into the database only once, corresponding with the more severe episode.

### 2.5. Clinical Evaluation at Hospital Admission and during Hospital Stay

Patient comorbidities and diagnosis codes were obtained from the electronic medical data records and classified according to the International Classification of Diseases, Ninth Revision, Clinical Modification (ICD-9-CM). During hospital stay, the data of six electronic health databases, namely, vital signs, laboratory, pharmacy prescription, interventional radiology, interventional cardiology and surgery, were integrated together using the number of the admission episode, which is unique for each patient and common to all these databases. Overall, the information extracted from these databases included: hemoglobin levels, leukocyte count, oxygen saturation, body temperature, blood pressure, heart rate and respiratory rate as well as a complete list of nephrotoxic drugs (detailed in Appendix A), and exposure to contrast dyes or major surgery. Every 24 h, updated information of all these data was dumped into the general study database which contained as well the comorbidity data and all available values of serum creatinine of each patient. From these data, a software generated classification codes for anemia, hypoxemic acute respiratory failure, Systemic Inflammatory Response Syndrome, shock, exposure to nephrotoxic drugs, contrast dyes or major surgery. Using these codes, the exposure to all these risk factors was classified as positive = 1, when the system detected at least one exposure during hospital stay, or negative = 0 when no exposure was detected. In all cases, the system recorded the data of exposure to each and one of these variables as well as the number of exposures to them. In patients with a code of AKI = 1, the exposure to these risk factors only was classified as =1 when it occurred within a maximum period of time prior to HA-AKI stage 3 detection (48 h for anemia, SIRS and shock, 72 h for contrast dyes and surgery and 7 days for nephrotoxic drugs). The procedures for the interrelation among the different electronic databases carried out to obtain the information on the clinical variables along hospital stay have been detailed in a previous report [28]. Unlike the hemoglobin level, arterial oxygen saturation, heart rate, respiratory rate or blood pressure level, that being numerical variables could be directly transferred to the general database, both circulatory shock and SIRS are complex variables that, to be automatically detected using a software-guided detection code, required the integration of data from various electronic records and the definition of classification algorithms. In both cases, before using them in statistical analyses, we analyzed the accuracy of the automatic detection systems in a sample of 3426 patients, as previously detailed [28].

### 2.6. Validation Set

The predictive model obtained at study set was externally validated in patients admitted at Arnau de Vilanova Hospital of Lleida between June 2017 and December 2019. Arnau de Vilanova hospital is a high-complexity teaching center and provides assistance to 490,000 habitants. This center develops similar activities as the study set with the exceptions of transplant programs and cardiac surgery. The selection of patients and the study procedures were done according to the same criteria stated for the study set. The external validation study was performed by an independent research team that did not participate in the development of the predictive model.

### 2.7. Statistics 

The incidence and prevalence calculations were referred the total number of admissions. For patients who developed more than one AKI episode along hospital admission, only the most severe episode was included in the study. Patients were considered to be at risk each time they were admitted to the hospital and, therefore, patients who during the study period were admitted two or more times were included in the calculations on each admission, except when readmission occurred within the 30 days after hospital discharge. Results are given as the mean ± SD or median and [P_25_–P_75_]. Differences in risk factors between groups were calculated by the Student’s unpaired T or ANOVA tests. Qualitative variables were compared using the Chi-squared test. Concordance analyses between qualitative variables was done by the Kappa coefficient. A *p* value of less than 0.05 was considered statistically significant. To determine which variables were independently associated with AKI, we carried out a univariate analysis comparing patients with and without AKI. All the variables with *p* values under 0.1 in the univariate analysis were entered into stepwise multiple logistic regression analysis with a forward selection method based on changes in the likelihood ratio (LR). Odds ratios (OR) were calculated from the regression coefficients as an approximation of the relative risk. The predictive value of the logistic model was evaluated using the C statistic, Cox & Snell R^2^ and Nagelgerkes’ R^2^. Model over-fitting was prevented using the Akaike Information Criterion (AIC) [30,31]. The Hosmer–Lemeshow’s test [32] was used as well to calculate the discrimination power and goodness of fit of the logistic model. Results are presented according to the TRIPOD guidelines for risk-prediction models [33,34]. Once obtained in the study set, the predictive logistic model was blindly tested on the external validation set by an independent group of researchers who did not participate in the development of the predictive model. Statistical analyses were performed with the Statistical Package for the Social Sciences for Windows 20.0.

## 3. Results

### 3.1. Study Set

Along the study period, there were 192,435 hospital discharges. Figure 1 shows the chart flow for patient selection. The final study group comprised 165,893 patients. Out of this cohort, 995 (0.60 %) developed HA-AKI stage 3.

Table 1 summarizes the demographic characteristics, comorbidities, clinical events and procedures along hospital stay in the study group, classified according to the presence of HA-AKI stage 3. HA-AKI stage 3 patients were older and more frequently male than non-AKI patients. Comorbidities including diabetes, hypertension, ischemic heart disease, ischemic peripheral vascular disease, chronic liver disease, chronic congestive heart failure, chronic obstructive pulmonary disease, malignancy, urologic disease and chronic kidney disease stages were also more frequent in AKI patients. The AKI risk increased linearly as glomerular filtration decreased. Patients with HA-AKI stage 3 showed also significantly higher rates of urgent admission, anaemia, acute respiratory failure, SIRS, shock, major surgery, and exposure to contrast dyes and to nephrotoxic drugs. 

The results of the logistic model to predict HA-AKI stage 3 are summarized in Table 2. The variables that had the strongest association with HA-AKI stage 3 were stage 3 of chronic kidney disease, diabetes mellitus and urological diseases, among chronic comorbidities, shock, acute respiratory failure, shock and urgent admission status, among acute complications, and major surgical procedures among the procedures performed. 

The model showed an AUC of 0.906 (95% CI 0.904 to 0.908), with a sensitivity of 89.1 (95% CI 87.0–91.0) and a specificity of 80.5 (95% CI 80.2–80.7) to predict HA-AKI stage 3 and showed an adequate calibration for high- and medium-risk categories but over-estimated the risk for low-risk categories. Table 3 (Chi^2^: 16.4, *p*: 0.034). 

The results of the stepwise forward procedures done to develop the final logistic model, including changes in the likelihood ratios, Cox and Snell R^2^, Nagelkerke R^2^ and AIC are summarized in the previous report [28].

### 3.2. Validation SET 

Along the study period there were 49,971 hospital discharges. Figure 2 shows the chart flow for patient selection. The final validation group comprised 43,569 patients.

The demographic characteristics, comorbidities and clinical parameters of the study and external validation cohorts are summarized in Table 4. 

When compared with the study set, patients of the validation set showed significantly lower prevalence of ischemic heart disease, ischemic cerebrovascular disease, chronic congestive heart failure, liver disease and major surgery. There was as well a significant difference in the distribution of chronic kidney disease stages between the two centers. In the validation set, 270 (0.62%) developed HA-AKI stage 3, with no significant differences between the study set and validation set. When the predictive model was tested in the validation set, it showed an AUC of 0.861 (95% CI 0.859–0.863) with a sensitivity of 83.0 (95% CI 80.5–85.3) and a specificity of 76.5 (95% CI 76.2–76.8) to predict HA-AKI and an adequate goodness of fit for all risk categories (Chi^2^: 15.42, *p*: 0.052). Table 5. 

The AUC was significantly lower than that observed in the study. Difference between AUC 0.0449, SD 0.00404 (95% CI 0.036–0.052), z 11.107 and *p* < 0.001) (Figure 3).

## 4. Discussion

In our study, we integrated the information of six electronic health databases, commonly used in the clinical practice, and we were able to develop the first predictive dynamic model that allows to estimate accurately, in non-critically ill patients, the individual likelihood of suffering HA-AKI stage 3 at any time during hospital stay. The final logistic model included the demographic data and the patient’s chronic comorbidities as well as a set of risk factors related to the patients’ clinical status and to the exposure to major surgery, contrast media or nephrotoxic drugs along hospital stay. In univariable analysis, those who developed HA-AKI stage 3 tended to be older and male. With respect to chronic comorbidities, diabetes, hypertension, ischemic heart disease, ischemic peripheral vascular disease, chronic liver disease, chronic congestive heart failure, chronic obstructive pulmonary disease, malignancy, urologic and chronic kidney disease were significantly more prevalent in patients who developed HA-AKI stage 3. All clinical variables evaluated, namely, anaemia, acute respiratory failure, acute heart failure, SIRS, circulatory shock, major surgery, and exposure to nephrotoxic drugs and to contrast media, were more prevalent in patients who developed HA-AKI stage 3. This model showed a high sensitivity and specificity to predict HA-AKI stage 3 and showed an adequate calibration for all, except for the lowest-risk categories for which it tended to over-estimate slightly the risk. This misclassification, however, affected only a few numbers of patients located at the lowest-risk categories. When compared with those previously published so far [35,36], the main novelty of our model is that it is the first one that predicts accurately the likelihood of suffering HA-AKI stage 3 along the whole hospital stay in non-critically ill patients rather than predicting the occurrence of AKI, regardless of its stage. Hence, it allows to estimate the individual likelihood of suffering severe AKI during hospitalization. The prospective monitoring of clinical data, through integration and cross-talk between different electronic databases, allowed us to analyze the dynamic exposure to risk factors related to the clinical status of patients along hospital stay, such as hypoxemia, hemoglobin level, blood pressure changes, contrast dyes or nephrotoxic drugs, prior to the detection of the HA-AKI stage 3 episode. This integration allowed as well to perform an accurate and reliable transformation of single variables such as blood pressure, heart rate, arterial oxygen saturation, prescription of vasoactive drugs or blood leukocyte counts into more complex variables defining specific syndromes such as SIRS and circulatory shock. Electronic records also permitted us to record the exposure to the same variables and risk factors in patients who did not develop HA-AKI stage 3 during hospital admission. This approach made it possible to estimate the individual risk, based on the actual exposure to each and one of risk factors. Since our predictive model was developed from the values of risk factors assessed prior to HA-AKI stage 3 detection, it allows to perform a dynamic monitoring of risk and even to predict the changes in the individual risk that are expected to happen every time the value of the different predictive risk factors changes. In order to obtain a predictive model that could be exportable to hospitals with different case-mix, patients who were admitted for programs and/or procedures such as cardiac surgery, solid organ or bone marrow transplantation, that are not commonly available to all hospital centers, were deliberately excluded from the study set. When comparing the study and the validation sets, we still observed statistically significant differences in the prevalence of several chronic comorbidities, in spite of the fact that, in both cohorts, we used the same ICD-9 codes to classy them. These differences may be due to dissimilarities in the case mix between both hospitals, but may also be caused by biases associated with potential discrepancies in assigning administrative codes to clinical conditions [37]. There were also between-group differences in other variables involved in the calculation of the risk of HA-AKI stage 3, such as the total percentage of urgent or surgical admissions. The discrimination ability of the model in the validation cohort was slightly but significantly lower than that observed in the original cohort. These differences are expected to be found when a predictive model is externally validated, and may be partially attributable to some degree of overfitting of the derivation modeling [38,39]. The calibration of the model in the external validation cohort showed a similar trend to that observed in the derivation cohort. Overall, the differences in the performance of the model between the study set and the validation set were small, which supports the potential scalability of the predictive model to fewer complex centers.

Our model has some limitations that must be highlighted. First, the record of clinical variables such as blood pressure, heart rate, respiratory rate or oxygen saturation were automatically dumped into the study database; however, these values are not without potential error related to the variability in the manual introduction of these variables into their corresponding databases. Second, the model obtained in our study is not the only one that can be obtained with the combination of data obtained from electronic records. As exposure to each of the acute complications or nephrotoxic agents can occur at different times after hospital admission, in order to relate the exposure to them with the development of HA-AKI stage 3, it was necessary to define a maximum period of time between exposure and detection of HA-AKI stage 3. In our study, the duration of this period of time was defined by consensus of the research group, using pathophysiological criteria. The definition of other periods of time, based on alternative criteria, would modify the prevalence of exposure to these risk factors and, consequently, the magnitude of the associations found between these variables and HA-AKI stage 3.

In conclusion, our study provides the first model, based on demographic data, specific comorbidities, acute clinical conditions and procedures, that can be used in clinical practice to obtain an accurate dynamic assessment of the individual risk of suffering HA-AKI stage 3 along the whole hospital stay period in patients admitted into non-critical hospitalization wards. This model allows from performing a repeated manual risk estimation, using the prediction algorithm, to providing an automatic risk measure updated in real time, in those centers where it is possible to carry out a complete integration among the health databases containing the necessary information. We anticipate that our study sets the cornerstone to a change in the management of hospital acute renal failure, by using a dynamic model of integration of electronic records with the aim of awareness of the physician in charge to these patients at high risk for AKI 3. It should be the aim to take special care to these patients at high risk to prevent acute renal failure and thus avoid fatal outcomes.

The anonymized database is available for reproduction as long as the requestor attaches a document endorsed by an ethical committee.

## Figures and Tables

**Figure 1 jcm-10-03959-f001:**
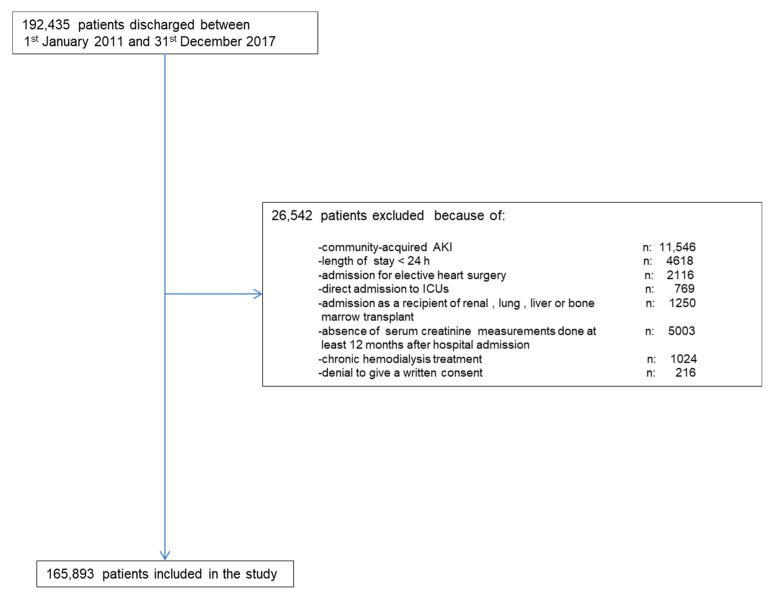
Flow-chart for patient’s selection.

**Figure 2 jcm-10-03959-f002:**
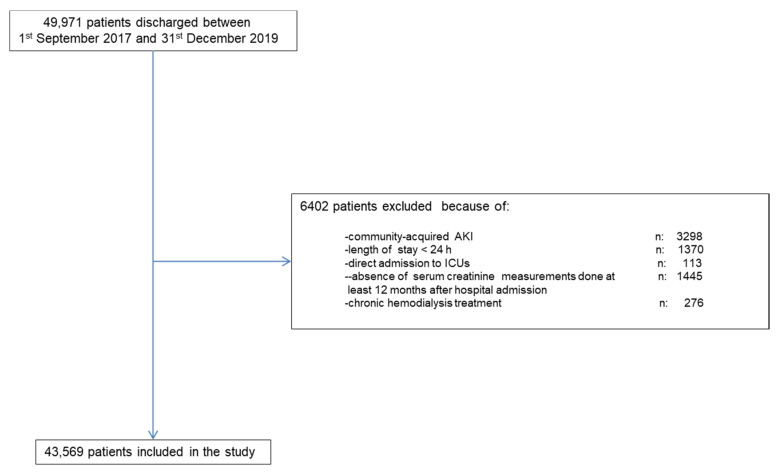
Shows the chart flow for patient selection.

**Figure 3 jcm-10-03959-f003:**
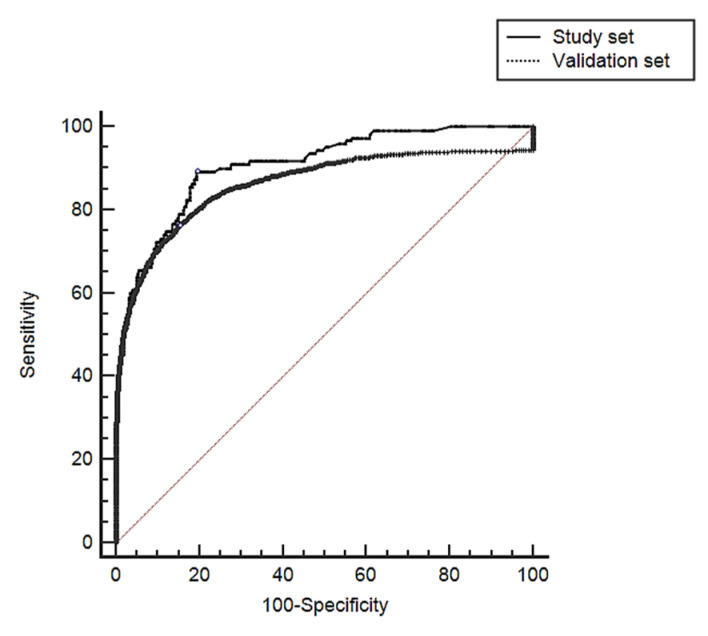
Comparison between AUCs obtained in the study set and in the validation set.

**Table 1 jcm-10-03959-t001:** Demographic characteristics, chronic comorbidities, clinical events and procedures along hospital admission, and univariate analysis of variables associated with HA-AKI stage 3 in the study group.

Variables	Total	Stage 3 AKI	Non-Stage 3 AKI	Sig
*n*	165,893	995 (0.6)	164,898 (99.4)	
Gender: Men. (*n*) %	74,962 (45.2)	517 (52.0)	74,445 (45.1)	<0.001
Age (years). mean (SD)	54.9 (20.6)	67.1 (21)	53.9 (19.9)	<0.001
Chronic comorbidities				
Diabetes. (*n*) %	30,357 (18.3)	450 (45.2)	29,907 (18.1)	<0.001
Hypertension. (*n*) %	65,554 (39.5)	707 (71.1)	64,847 (39.3)	<0.001
Ischemic Heart Disease. (*n*) %	12,428 (7.5)	169 (17.1)	12,259 (7.4)	<0.001
Ischemic Cerebrovascular disease. (*n*) %	11,446 (6.9)	78 (7.8)	11,368 (6.9)	0.136
Ischemic Peripheral vascular disease. (n) %	8706 (5.2)	93 (9.3)	8613 (5.2)	<0.001
Chronic digestive disease. (*n*) %	9627 (5.8)	51 (5.1)	9576 (5.8)	0.198
Chronic liver disease. (*n*) %	5667 (3.4)	105 (10.6)	5562 (3.4)	<0.001
Chronic congestive heart failure. (*n*) %	14,344 (8.6)	256 (25.7)	14,088 (8.5)	<0.001
Chronic obstructive pulmonary disease. (*n*) %	23,272 (14.0)	424 (42.6)	22,848 (13.9)	<0.001
Malignancy. (*n*) %	23,504 (14.2)	304 (30.6)	23,200 (14.1)	<0.001
Rheumatologic disease. (*n*) %	6828 (4.1)	41 (4.1)	6787 (4.1)	0.529
Urologic disease. (*n*) %	11,926 (7.2)	148 (14.9)	11,778 (7.1)	<0.001
Chronic Kidney disease stages				<0.001
0 + I	137,385 (82.8)	583 (58.6)	136,802 (83)	
II	16,252 (9.8)	109 (11.0)	16,143 (9.8)	
III	9265 (5.6)	175 (17.6)	9090 (5.5)	
IV	2991 (1.8)	128 (12.9)	2863 (1.7)	
Clinical variables along hospital admission				
Urgent admission. (*n*) %	108,577 (65.5)	947 (95.2)	107,630 (65.3)	<0.001
Anaemia. (*n*) %	23,291 (14.0)	379 (38.1)	22,912 (13.9)	<0.001
Acute respiratory failure. (*n*) %	7803 (4.7)	308 (31.0)	7495 (4.5)	<0.001
Acute Hearth failure (*n) %*	6204 (3.7)	241 (24.2)	5963 (3.6)	<0.001
SIRS. (*n*) %	2358 (1.4)	235 (23.6)	2123 (1.3)	<0.001
Circulatory shock. (*n*) %	2018 (1.2)	280 (28.1)	1738 (1.1)	<0.001
Major surgery. (*n*) %	61,583 (37.1)	408 (41.0)	61,675 (37.4)	<0.001
Exposure to contrast media. (*n*) %	14,698 (8.9)	280 (28.1)	14,418 (8.7)	<0.001
Exposure to nephrotoxic drugs. (*n*) %	85,863 (51.8)	677 (68.0)	85,186 (51.7)	<0.001

**Table 2 jcm-10-03959-t002:** Variables independently associated with HA-AKI stage 3 in the logistic regression analysis.

Variable	B	S.E.	Wald	OR	95% CI	*p*-Value
Age	0.024	0.003	91.2	1.03	1.02–1.03	0.000
Hypertension	0.539	0.084	41.1	1.71	1.45–2.02	0.000
Diabetes	1.184	0.079	223.5	3.27	2.79–3.81	0.000
Peripheral vascular disease	0.597	0.135	19.7	1.82	1.39–2.37	0.000
Anaemia	0.664	0.075	78.0	1.94	1.67–2.25	0.000
Chronic congestive hearth failure	0.405	0.085	22.5	1.50	1.27–1.77	0.000
Ischemic hearth disease	0.653	0.107	37.6	1.92	1.56–2.37	0.000
Chronic obstructive pulmonary disease	0.469	0.096	23.9	1.60	1.32–1.93	0.000
Chronic liver disease	1.013	0.133	58.1	2.75	2.12–3.57	0.000
Chronic urologic disease	1.309	0.118	123.9	3.70	2.94–4.66	0.000
CKD_stage			469.9			0.000
CKD_stage(1)	0.582	0.122	22.7	1.79	1.41–2.27	0.000
CKD_stage(2)	1.425	0.1	204.0	4.16	3.49–5.05	0.000
CKD_stage(3)	2.187	0.119	339.8	8.91	7.06–11.24	0.000
SIRS	0.698	0.128	29.6	2.01	1.56–2.59	0.000
Shock	2.055	0.122	286.1	7.81	6.15–9.9	0.000
Acute Hearth Failure	0.801	0.096	69.9	2.23	1.84–2.69	0.000
Major_surgery	1.213	0.083	211.8	3.36	2.85–3.96	0.000
Acute respiratory failure	1.283	0.106	147.4	3.61	2.93–4.44	0.000
Nephrotoxic drugs	0.345	0.078	19.8	1.41	1.21–1.64	0.000
Exposure to contrast dyes	0.931	0.085	119.5	2,53	2.15–2.99	0.000
Urgent_admission	1.899	0.161	139.0	6.68	4.87–9.15	0.000
Constant	−11.211	0.237	2239.0	0.00		

**Table 3 jcm-10-03959-t003:** Hosmer and Lemeshow’s goodness of fit of the logistic predictive model in the study group.

Risk Deciles	Acute Kidney Injury = 0		Acute Kidney INJURY = 1		Total
	Observed	Expected	Observed	Expected	
<0.0001702	16,514	16,512.6	0	1.4	16,514
0.0001702–0.0003350	16,587	16,586.0	2	3.0	16,589
0.0003351–0.0004798	16,584	16,580.2	1	4.8	16,585
0.0004799–0.0007357	16,577	16,581.0	12	8.0	16,589
0.0007358–0.0011664	16,549	16,558.2	22	12.8	16,571
0.0011665–0.0016138	16,556	16,554.0	19	21.0	16,575
0.0016139–0.0027321	16,557	16,554.9	32	34.1	16,589
0.0027322–0.0044384	16,527	16,526.5	56	56.5	16,583
0.0044385–0.0098603	16,463	16,478.8	126	110.2	16,589
>0.0098603	15,984	15,965.7	725	743.3	16,709

Chi-square: 16.4, *p*: 0.034.

**Table 4 jcm-10-03959-t004:** Comparison of demographic characteristics, comorbidities and clinical variables between the study set and the external validation set.

Variables	Study Set	Validation Set	*p*-Value
*n*	165,893	43,569	
HA-AKI Stage 3	995 (0.60)	271 (0.62)	0.594
Gender: Men. (*n*) %	74,962 (45.2)	19,606 (44.9)	0.105
Age (years). mean (SD)	54.9 (20.6)	55.7 (22.1)	0.389
Chronic comorbidities			
Diabetes (*n*) %	30,357 (18.3)	7840 (17.9)	0.048
Hypertension (*n*) %	65,554 (39.5)	16,991 (38.9)	0.059
Ischemic Heart Disease (*n*) %	12,428 (7.5)	3033 (6.9)	<0.001
Ischemic Cerebrovascular disease (*n*) %	11,446 (6.9)	2614 (6.0)	<0.001
Ischemic Peripheral vascular disease (*n*) %	8706 (5.2)	2396 (5.5)	0.037
Chronic digestive disease (*n*) %	9627 (5.8)	2483 (5.7)	0.407
Chronic liver disease (*n*) %	5667 (3.4)	1307 (3.0)	<0.001
Chronic congestive heart failure (*n*) %	14,344 (8.6)	3267 (7.5)	<0.001
Chronic obstructive pulmonary disease (*n*) %	23,272 (14)	6535 (15.0)	<0.001
Malignancy (*n*) %	23,504 (14.2)	6317 (14.5)	0.081
Rheumatologic disease (*n*) %	6828 (4.1)	1743 (4.0)	0.285
Urologic disease *(n*) %	11,926 (7.2)	3135 (7.1)	0.971
Chronic Kidney Disease stages			0.2758
0 + I	137,385 (82.8)	36,162 (83.0)	
II	16,252 (9.8)	4182 (9.6)	
III	9265 (5.6)	2396 (5.5)	
IV	2991 (1.8)	829 (1.9)	
Clinical variables along hospital admission			
Urgent admission (*n*) %	108,577 (65.5)	28,319 (65.0)	0.077
Anaemia (*n*) %	23,291 (14.0)	6186 (14.2)	0.397
Acute respiratory failure (*n*) %	7803 (4.7)	2178 (5.0)	0.011
Acute Hearth failure (*n*) %	6204 (3.7)	1655 (3.8)	0.565
SIRS (*n*) %	2358 (1.4)	653 (1.5)	0.227
Circulatory shock (*n*) %	2018 (1.2)	566 (1.3)	0.167
Major surgery (*n*) %	61,583 (37.1)	13,942 (32.0)	<0.001
Exposure to contrast dyes (*n*) %	14,698 (8.9)	3.921 (9.0)	0.36
Exposure to nephrotoxic drugs (*n*) %	85,863 (51.8)	23,135 (53.1)	<0.001

**Table 5 jcm-10-03959-t005:** Hosmer and Lemeshow’s goodness of fit of the logistic predictive model in the validation group.

	Acute Kidney Injury = 0		Acute Kidney Injury = 1		Total
Risk Deciles	Observed	Expected	Observed	Expected	
<0.0001486	4342	4343.4	2	0.58	4344
0.0001486–0.0002375	4347	4347.7	2	1.30	4349
0.0002376–0.0003818	4374	4371.8	0	2.12	4374
0.0003819–0.0006162	4353	4355.7	6	3.23	4359
0.0006163–0.0009573	4351	4352.3	6	4.70	4357
0.0009574–0.0015601	4345	4351.2	9	6.74	4358
0.0015602–0.0025301	4347	4345.1	8	9.86	4355
0.0025302–0.0044511	4349	4341.6	8	15.3	4357
0.0044512–0.0101964	4327	4329.1	31	28.8	4358
>0.0101964	4159	4157.7	199	200.24	4358

Chi-square: 15.416, *p*: 0.052.

## Data Availability

The anonymized database is available for reproduction as long as the requestor attaches a document endorsed by an ethical committee.

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
