# Peer review of "Development and Validation of a Model to Predict Severe Hospital-Acquired Acute Kidney Injury in Non-Critically Ill Patients"

_jcm, 2021, doi:10.3390/jcm10173959_

Round 1

Reviewer 1 Report

The authors investigated the incidence of hospital associated acute kidney injury (HA-AKI). They focused on the several risk factors of development of HA-AKI stage 3 in non-critically ill hospital patients. They described that a multivariate model to predict HA-AKI stage3 can be used in clinical practice to obtain an accurate dynamic assessment of the individual risk of HA-AKI stage 3 in non-critically ill patients. The Methods section is clear. The data is presented clearly. These findings will be of interest to clinical practitioners, as well as researchers in the field.

Major comments

I have the following concerns.

The authors demonstrated the detailed data to predict the development of AKI, including several chronic comorbidities, clinical variables along hospital admission, and gender/age. The model with these data had high specificity and sensitivity to HA-AKI stage 3. However, the model itself is not useful to prevent the development of HA-AKI. I think that it is crucial to develop a scoring system to predict the incidence of HA-AKI using this data in non-critically ill patients.

Minor comments

In the results section, the authors state that “(Chi2: 16.4, p:0.034) on line 236, while Table 3 showed “Chi-square: 16.4, p: 0.0034.”. Which is correct?

The manuscript is improved as a result of the revisions, and the authors have adequately addressed most of the comments from the original review. There is just one point that the authors have not properly or fully addressed, noted below.

Author Response

Response to Reviewer 1 Comments

Major comments

I have the following concerns.

The authors demonstrated the detailed data to predict the development of AKI, including several chronic comorbidities, clinical variables along hospital admission, and gender/age. The model with these data had high specificity and sensitivity to HA-AKI stage 3. However, the model itself is not useful to prevent the development of HA-AKI. I think that it is crucial to develop a scoring system to predict the incidence of HA-AKI using this data in non-critically ill patients.

 Response 1:

Our model is a logistic model that calculates the individual probability of presenting HA-AKI, which is the basis for predictive models to be developed on this probability. It is not intended for a predictive purpose, but for risk estimation. Our group is currently working on the development of prevention models.

Minor comments

In the results section, the authors state that “(Chi2: 16.4, p:0.034) on line 236, while Table 3 showed “Chi-square: 16.4, p: 0.0034.”. Which is correct?

Response 2:

There was a writing error. The correct is: Chi2: 16.4, p:0.034

Reviewer 2 Report

Review:

The manuscript of Jacqueline Del Carpio, et al. studies the epidemiology of acute renal failure in patients admitted to conventional hospitalization wards. This is interesting; the incidence of hospital-acquired AKI (HA-AKI) is much less known for patients admitted to conventional hospitalization wards than patients admitted to the intensive care unit. The study provides a model to obtain an accurate dynamic assessment of the individual risk of HA-AKI stage 3 along the hospital stay period in non-critically ill patients, means not admitted to intensive care unit.

Methods: How did the authors manage to get AKI along the whole hospital stay period in patients admitted into non-critical hospitalization wards? Specifically, how fluctuating AKI is detected and handled in the analyses and the model. For example, a patients that has developed two or three AKI3 during his stay in the hospital. In a tertiary hospital, this is frequent and should be explained, or noted as limit of the study. In methods, it is stated that “The number of the admission episode was used as a filter so that patients with more than one HA-AKI stage 3 episode during hospital stay were entered into the data base only once, corresponding with the more severe episode.” Is this correct? Please explain the rational – and limitation of the design -  in discussions action. And state how many patients had more than one episode of AKI 3 in the hospital stay, how many 2, 3, 4,.. episodes.

AKI should be spelled out once, not use the abbreviation since the beginning, even in title. I would suggest to spell it out in the title and then again at the beginning of the Introduction, where it is appropriate.

Title:

Development and external validation: Please delete “external” from “external validation”. It makes no sense in the title, and it is not critical to the study.

It is confusing to see so many authors coming from quite different hospitals and institutions and so many have more than one affiliations. Second, difficult to understand that the study was done in one hospital, “included patients admitted to hospital Vall d’Hebron”, and so many affiliations. Please explain exactly the role and the contribution of each author and its affiliations. The meaning of the (+) of the three last authors is not explained in a footnote etc. Please precise. I do not think that there can be three last authors contribution equally. Please explain. And reduce to two equally contributing last authors maximum. If there have to be three last contributing authors, the editor should be consulted.

Methods, line 101: Please delete “high-complexity “ in “Vall d’Hebron is a “tertiary high-complexity hospital”. Per definition, a tertiary hospital is “high-complexity”.

Table 2: please correct “Chronic cingestive hearth failure” to “Chronic congestive heart failure”.

End document:

Please fill up lines 357 to 362.

Author Contributions: 357

Funding: 358

Institutional Review Board Statement: 359

Informed Consent Statement: 360

Data Availability Statement: 361

Conflicts of Interest: The authors declare no conflict of interest 362

It is hard to believe in our time, not to say unbelievable, with so many authors, that in a “tertiary high-complexity hospital” there is no conflict of interest. Please fill it up correctly.

Acknowledgements are missing. This is also hard to believe in a “tertiary high-complexity hospital” of today. Please fill it out.

The conclusion is interesting and well written, it can be concluded that the study anticipates, but only anticipates, it has to be shown first that awareness of the physician leads to clinical improvement. I would suggest modifying slightly to: “We anticipate that our study sets the cornerstone to a change in the management of hospital acute renal failure, by using a dynamic model of integration of electronic records with the aim of awareness of the physician in charge to these patients at high risk for AKI 3. It should be the aim to take special care to these paitents at high risk to preventing acute renal failure and thus avoiding fatal outcomes.”

Author Response

Response to Reviewer 2 Comments

Review:

The manuscript of Jacqueline Del Carpio, et al. studies the epidemiology of acute renal failure in patients admitted to conventional hospitalization wards. This is interesting; the incidence of hospital-acquired AKI (HA-AKI) is much less known for patients admitted to conventional hospitalization wards than patients admitted to the intensive care unit. The study provides a model to obtain an accurate dynamic assessment of the individual risk of HA-AKI stage 3 along the hospital stay period in non-critically ill patients, means not admitted to intensive care unit.

Methods: How did the authors manage to get AKI along the whole hospital stay period in patients admitted into non-critical hospitalization wards? Specifically, how fluctuating AKI is detected and handled in the analyses and the model. For example, a patients that has developed two or three AKI3 during his stay in the hospital. In a tertiary hospital, this is frequent and should be explained, or noted as limit of the study. In methods, it is stated that “The number of the admission episode was used as a filter so that patients with more than one HA-AKI stage 3 episode during hospital stay were entered into the data base only once, corresponding with the more severe episode.” Is this correct? Please explain the rational – and limitation of the design -  in discussions action. And state how many patients had more than one episode of AKI 3 in the hospital stay, how many 2, 3, 4,.. episodes.

Response 1: Yes, were entered into the data base only once, corresponding with the more severe episode. Our measure of incidence is episode per patient admitted, not by number of hospitalizations. This approach effectively supposes a limitation because it supposes excluding the variable individual susceptibility to present AKI. Our group is working in a new model to look for the profile of the patient likely to present several AKI episodes.

How many patients had more than one episode of AKI 3 in the hospital stay, how many 2, 3, 4,..episodes?

We do not fear this data because only the most serious episode of AKI was entered into the data base.

AKI should be spelled out once, not use the abbreviation since the beginning, even in title. I would suggest to spell it out in the title and then again at the beginning of the Introduction, where it is appropriate.

Response 2: Change done

Title:

Development and external validation: Please delete “external” from “external validation”. It makes no sense in the title, and it is not critical to the study.

Response 3: Change done

It is confusing to see so many authors coming from quite different hospitals and institutions and so many have more than one affiliations. Second, difficult to understand that the study was done in one hospital, “included patients admitted to hospital Vall d’Hebron”, and so many affiliations. Please explain exactly the role and the contribution of each author and its affiliations. The meaning of the (+) of the three last authors is not explained in a footnote etc. Please precise. I do not think that there can be three last authors contribution equally. Please explain. And reduce to two equally contributing last authors maximum. If there have to be three last contributing authors, the editor should be consulted.

Response 4:

Our study was carried out in 2 centers (Vall d’Hebron and Arnau of Vilanova hospital). The four cores of the study are Vall d’Hebron hospital, Arnau de Vilanova hospital, Catalonian Institute of health and Amalfi analytics.

Some authors have more than one affiliation because during the time that the study has lasted, they have changed their workplace. To avoid confusion, we have deleted an affiliation (3) that was not necessary for this publication.

The meaning of the (+): Principal investigator.

Methods, line 101: Please delete “high-complexity “in “Vall d’Hebron is a “tertiary high-complexity hospital”. Per definition, a tertiary hospital is “high-complexity”.

Response 5: Change done

Table 2: please correct “Chronic cingestive hearth failure” to “Chronic congestive heart failure”.

Response 6: Change Done

It is hard to believe in our time, not to say unbelievable, with so many authors, that in a “tertiary high-complexity hospital” there is no conflict of interest. Please fill it up correctly.

Response 7: After re-commenting with each of the authors, all of them declare that they have no conflict of interest that falls within the objectives of this article.

Acknowledgements are missing. This is also hard to believe in a “tertiary high-complexity hospital” of today. Please fill it out.

Response 8:

We thank the entire team of the nephrology, internal medicine, emergency department, nursing team and computer scientists who collaborated with the realization of this manuscript.

The conclusion is interesting and well written, it can be concluded that the study anticipates, but only anticipates, it has to be shown first that awareness of the physician leads to clinical improvement. I would suggest modifying slightly to: “We anticipate that our study sets the cornerstone to a change in the management of hospital acute renal failure, by using a dynamic model of integration of electronic records with the aim of awareness of the physician in charge to these patients at high risk for AKI 3. It should be the aim to take special care to these patients at high risk to preventing acute renal failure and thus avoiding fatal outcomes.”

Response 9: Change done